# High Prevalence of Anemia and Poor Compliance with Preventive Strategies among Pregnant Women in Mwanza City, Northwest Tanzania: A Hospital-Based Cross-Sectional Study

**DOI:** 10.3390/nu14183850

**Published:** 2022-09-17

**Authors:** Eveline T. Konje, Bernadin Vicent Ngaila, Albert Kihunrwa, Stella Mugassa, Namanya Basinda, Deborah Dewey

**Affiliations:** 1Department of Biostatistics and Epidemiology, School of Public Health, Catholic University of Health and Allied Sciences—BUGANDO, Mwanza P.O. Box 1464, Tanzania; 2Department of Obstetrics and Gynecology, Weill Bugando School of Medicine, Catholic University of Health and Allied Sciences—BUGANDO, Mwanza P.O. Box 1464, Tanzania; 3Department of Community Medicine, School of Public Health, Catholic University of Health and Allied Sciences—BUGANDO, Mwanza P.O. Box 1464, Tanzania; 4Departments of Pediatrics and Community Health Sciences, Cumming School of Medicine, University of Calgary, Calgary, AB T2N 1N4, Canada; 5Owerko Centre at the Alberta Children’s Hospital Research Institute, University of Calgary, Calgary, AB T2N 1N4, Canada

**Keywords:** anemia preventive strategies, anemia in pregnancy, compliance with strategies

## Abstract

Anemia in pregnancy is prevalent in Tanzania despite the implementation of existing prevention strategies. This study aims to determine the level of compliance with anemia preventive strategies among pregnant women and the factors associated with poor compliance. A cross sectional study was conducted among 768 pregnant women who attended the Bugando Medical Center, Sekou-Toure Regional Hospital, Nyamagana District Hospital, and Buzuruga Health Center in Mwanza, Northwest Tanzania. The prevalence of anemia at term was 68.8% (95% CI, 65.5–72.0%). The average hemoglobin level at term was 10.0 g/dL (95% CI, 9.8–10.1). Only 10.9% of pregnant women complied fully with anemia-preventive strategies. A decrease in mean hemoglobin level was observed across levels of compliance, with women who were non-compliant displaying a significantly lower mean hemoglobin level (8.3 g/dL) compared to women who were fully compliant (11.0 g/dL). Poor compliance was associated with no formal or primary education and initiating antenatal care in the 2nd or 3rd trimester. Anemia in pregnancy was commonly associated with lack of compliance with preventive strategies among participants. There is a need for community-based health education on the importance of complying with anemia-preventive strategies in order to reduce the burden during pregnancy and the consequences of anemia to the unborn baby.

## 1. Introduction

Anemia in pregnancy is prevalent in low- and middle-income countries (LMICs) despite the existence of anemia-preventive strategies such as the use of insecticide treated nets, anti-malaria prophylaxis, and preventive anthelminthic treatments and hematinics [1,2,3,4]. In 2011, it was estimated that worldwide, 38% of pregnant women were anemic [2,4,5], and in Tanzania, more than a half of pregnant women (57%) have been found to be anemic [6]. The cause of anemia is multifactorial, including iron and folate deficiency, chronic infections, and helminth infestations [1,3]. Severe anemia in pregnancy can have negative consequences for both the mother and infant. Common maternal complications include maternal fatigue, poor cognitive performance, cardiovascular disease, and even maternal death [7]. Complications to the infant include premature birth, low birth weight, intrauterine fetal growth restriction, and fetal death [7,8,9,10]. Maternal anemia has also been associated with negative long-term health consequences in children, such as poor growth, increased cardiovascular disease risk, poor cognitive and motor development, obesity, and depression [7,11,12].

To reduce and prevent anemia in pregnancy in LMICs, the WHO guidelines recommend iron supplementation, intermittent malaria treatment with sulfadoxine-pyrimethamine (SP), use of treated mosquito nets, and deworming during antenatal care (ANC) [13,14]. Although inconsistent results have been reported on the effectiveness of these interventions on pregnancy outcomes and hemoglobin (Hb) concentrations, some studies support their effectiveness [13,15,16,17,18,19,20]. For example, several studies have reported a reduction in anemia among pregnant women who used oral iron for more than 90 days [1,9,16,17,19]. Further, deworming with an anthelminthic was associated with reduced anemia in pregnancy and adverse pregnancy outcomes [19]. The use of antimalarial prophylaxis with SP in malaria-endemic areas has also been associated with lower levels of anemia and fewer adverse pregnancy outcomes [18,21]. Despite the benefits of these anemia-preventive measures, anemia in pregnancy remains prevalent in developing countries [2,3,4,15].

A focused antenatal care (FANC) model that is currently implemented in Tanzania emphasizes that standard clinical practice for all pregnant women includes the prescription and provision of iron supplements, malaria prophylaxis, and anthelminthics. It is recommended that iron supplements (i.e., combined ferrous sulfate 200 mg with 60 mg elemental iron and 0.25 mg folic acid) be taken once or twice daily over the entire pregnancy. Antimalarial prophylaxis (i.e., three tablets of SP; 500 mg sulfadoxine/25 mg pyrimethamine per tablet) are to be taken orally once during the second trimester and sometime during the third trimester. A single dose of an anthelminthic, either albendazole (400 mg single dose) or mebendazole (500 mg single dose), is to be taken in the second and/or third trimester [22,23]. Lastly, pregnant women are to be provided with an insecticide-treated net (ITN) and are encouraged to sleep under it. These preventative strategies are to be provided freely at public and private health facilities as per current reproductive and child health guidelines. 

In Tanzania, it has been reported that 51% of pregnant women attend ANC clinics at least four times during pregnancy, and almost all women attend at least once. Notably, 57% of pregnant women have been found to be anemic at the end of their pregnancy [6]. Northwestern Tanzania is one of the top three regions of the country with the highest prevalence of anemia among pregnant women and the lowest proportion of pregnant women receiving ANC [6]. Further, a maternal mortality audit conducted in 2016 reported that anemia contributed to more than 15% of the maternal deaths that occurred at Bugando Medical Center (BMC), Sekou Toure Regional Hospital, and Nyamagana District Hospital [24], which are in Mwanza city, northwest Tanzania. The effectiveness of the anemia-preventive strategies in reducing maternal anemia depends upon the timing of the intervention, adherence to the intervention, and the existence of iron reserves at conception and at the time of ANC initiation [7,15,17]. Previous research has reported on maternal compliance with iron supplementation [25,26,27]. However, to the best of our knowledge, no study has examined women’s compliance with all four anemia-preventative measures. As the causes of anemia are multifactorial [1,3,5,15], investigating compliance with anemia prevention strategies in relation to the common causes of anemia (malaria, helminths, and iron deficiency) is important to inform preventive approaches. Hence, the aim of this study was to investigate the prevalence of anemia among pregnant women at term, their compliance with anemia-preventive strategies, and the factors associated with poor compliance. The study findings highlight the levels of compliance with anemia-preventive strategies among pregnant women, which could be used to identify gaps in antenatal care services. 

## 2. Materials and Methods

### 2.1. Study Design and Population

A facility-based cross-sectional study was conducted in antenatal wards of health facilities located in Mwanza City, Bugando Medical Center (BMC), Sekou-Toure Regional Hospital, Nyamagana District Hospital, and Buzuruga Health Center, over a five-month period (January–May 2017). Bugando Medical Centre serves as a consultancy and teaching hospital for regions in the Lake and Western Zones of Tanzania. Sekou-Toure is a regional referral hospital receiving patients from six districts of the Mwanza region. Nyamagana District Hospital receives patients from three different health centers, and Buzuruga Health Center serves the population from the Ilemela district in Mwanza city. Based on the health facility records in 2017, the average daily deliveries were 20 at BMC, 45 at Sekou Toure, 15 at Nyamagana, and 7 at Buzuruga. Antenatal wards in these study sites receive women who have pregnancy-related complications (pre-eclampsia, anemia, cardiomyopathy, women in first-stage labor waiting for progression, those who come for planned deliveries (previous c-section, placenta previa)) and other high-risk pregnancies. 

### 2.2. Sample Size and Sampling Procedure

The sample size was estimated using a Kish Leslie’s formula considering “z” for a 95% confidence interval, 5% tolerable error, and 57% anemia prevalence in pregnancy in Tanzania. A minimum of 377 participants was required. Due to differences in health facility levels (tertiary, regional, district, and health center), the sample size was inflated by a design effect of two. Hence, we recruited a total of 768 pregnant women at term. A proportion to size approach based on average daily deliveries was considered when recruiting participants from each site. Hence, different numbers of participants were recruited from BMC (*n* = 212), Sekou-Toure (*n* = 367), Nyamagana (*n* = 136), and Buzuruga (*n* = 53), making a total of 768 participants. Women at term (gestation age ≥ 37 weeks based on the date of last normal menstrual period) who were admitted to the antenatal ward but not in active labor were conveniently selected to participate in the study. Informed written consent was obtained before delivery when women were not in labor. Those with chronic diseases such as sickle cell anemia or diabetic mellitus, or emergency gynecological conditions were excluded from the study. During the study period, 5496 pregnant women were admitted for delivery in the four selected health facilities, but only those who met the inclusion criteria and consented to participate were involved in this study.

### 2.3. Hemoglobin Measurement

A capillary spot blood sample was collected from each participant. The tip of a finger was disinfected and pricked; the first drop of blood was discarded, and the second drop was collected and used to measure hemoglobin (Hb) level using a calibrated HemoCue hemoglobimeter system (Hemocue Hb 201^+^ system). This was used to measure Hb levels in all participants in the study and calibrated on a daily basis. For 10% of participants, two millimeters of venous blood was collected in an EDTA tube and transported to the BMC laboratory within two hours. These venous blood samples were used for quality control of Hb level estimations; the Hb level from spot blood and venous blood samples were compared in the same woman. In rare situations, when the discrepancy was considered unacceptable, the readings based on venous blood samples were recorded. Furthermore, the Hb level at the first ANC visit was extracted from women’s RCH-4 card, which is usually presented by women when they come to the health facility for childbirth care services. 

### 2.4. Description of Variables 

To assess compliance with anemia-preventive strategies the following four strategies were assessed: insecticide-treated net (ITN) use in the past three months (yes/no), antimalarial prophylaxis use of sulfadoxine-pyrimethamine, use of mebendazole/albendazole for deworming, and use of iron and folate supplements. Those who reported being provided with antimalarials were asked if they swallowed the pills as per the health providers’ instructions and how many times they were provided with and swallowed the antimalarials. We also asked similar questions regarding iron and folic acid supplements and mebendazole/albendazole (i.e., deworming medications) use. A composite variable was created to assess compliance with all four anemia-preventive strategies provided during antenatal care. Insecticide-treated net use was scored “1” for “yes” and “0” for “no”. The use of antimalarial was scored “1” for a complete dose (at least two doses) and “0” other (single dose or none). The use of mebendazole/albendazole for deworming was scored “1” for a complete dose (at least two doses) and “0” other (i.e., one does or none). Use of iron and folic acid supplements was scored “1” for women who reported taking these supplements for at least 90 days (i.e., at least three months) and “0” other (i.e., taking these supplements for less than 90 days or not at all). The total score on this composite variable ranged from 0 to 4, with higher scores indicating better compliance to the anemia-preventive strategies. We combined women with scores of 3 and 4 into one group for analyses purposes because only seven women in our sample received a score of 4. The classifications for compliance were as follows: a score of 3 or 4 was classified as “fully compliant”; a score of 2 was classified as “mainly compliant”; a score of 1 was classified as “somewhat compliant”; and a score of 0 was classified as “non-compliant”. Anemia in pregnancy was defined according to WHO as an Hb level less than 11 g/dL and severe anemia as an Hb level less than 7 g/dL. Since Hb level measurements are affected by altitude, all levels were adjusted for altitude before prevalence was estimated and mean Hb in g/dL was calculated.

### 2.5. Data Management and Analysis

Data analysis was completed using STATA version 13.0. Descriptive statistics are presented as proportions/percentages or means and standard deviations; mean Hb levels (g/dL) are presented with 95% confidence intervals (CI), and prevalence of anemia and level of compliance are presented as proportions. All categorical variables, such as education level, occupation, and anemia-preventive strategies (use of ITNs, antimalarial, deworming medications, and iron and folate supplements), were summarized using proportions. The differences in mean Hb levels (g/dL) across the levels of compliance were compared using ANOVA with Bonferroni corrections for multiple comparisons and Bartlett’s test for equal variances. Compliance was treated as an ordinal variable based on the categorization of the scores; a proportional odds model was used to assess factors associated with poor compliance after assessing for the parallel lines assumption using the parallel for all significant variables. Odds ratios were used as measures of association and reported with 95% confidence intervals. No adjustment for study site was performed since compliance levels were not statistically different across the study sites. The likelihood ratio test was used to compare the model with all variables and the reduced model for identification of confounding variables. All analyses were considered statistically significant at <0.05.

Previous studies have reported that pregnancy trimester, smoking, and altitude can inflate or deflate Hb levels [27,28]. In the present study, we adjusted Hb measurements for all participants who were in their third trimester by adding 1.0 g/dL to their individual hemoglobin value [28]. Hemoglobin concentration levels were not adjusted for smoking status because the smoking prevalence among participants was <0.1%. However, it was adjusted for altitude, as Mwanza is located at 1140 m above sea level, and the adjustment was carried out using Hb adjustment = −0.032 × (altitude × 0.0032808) + 0.022 × (altitude × 0.0032808)^2^ [28].

### 2.6. Sensitivity Analysis

We also conducted sensitivity analyses by examining the effects of changing the scoring criteria for compliance and examining if this affected the estimated mean Hb and prevalence of anemia across compliance levels. To explore the effect of using different cut-offs to classify compliance, we used the composite variable of interest (compliance) that was classified based on score of 3–4 as “mainly compliant”, a score of 2 as “somewhat compliant”, and scores of 1 or 0 as “noncompliant”. We also assessed the change in Hb level for each category of the compliance using paired *t*-tests for participants with Hb measures at ANC initiation and at term.

## 3. Results

### 3.1. Socio-Demographic and Obstetric Characteristics

Participants had a mean age of 25.5 years (SD = 5.7 years). Most lived in Ilemela (46.1%) and Nyamagana (44.1%) districts, which are the catchment areas for the selected health facilities. Of all participants, 84.4% (649/768) were married, 53.7% (412/768) reported having a primary education level, and 40.8% (313/768) were housewives. The number of pregnancies reported by women who participated in the study ranged from one to ten, with a median of two pregnancies. Almost 40 percent of participants were nulliparous (39.8%, 306/768). More than half of the participants (62.5%, 480/768) attended antenatal care (ANC) clinics at dispensaries (i.e., lowest level of health facility in the country). About two-thirds of the participants (68.5%, 526/768) initiated ANC in the second trimester; less than one-quarter (22.1%, 170/768) did so in the first trimester of pregnancy. Almost all the women attended ANC at least once (99.7%, 766/768) (see Table 1 below).

### 3.2. Anemia in Pregnancy at Time of ANC Initiation and at Term

Approximately, one-quarter of all participants (26.6%, 204/768) had no Hb measurement information recorded on their ANC cards although they attended ANC. Among those women who had Hb measurements recorded, the prevalence of anemia was 50.5% at ANC initiation (95% CI 46.4–54.7%). The prevalence of anemia at term before and after adjusting for altitude and 3rd trimester was 67.2% (95% CI 63.9–70.5%) and 68.8% (95% CI, 65.5–72.0%), respectively. Mean Hb at term was 10.1 g/dL (95% CI, 9.9–10.2), and adjusted mean Hb at term was 10.0 g/dL (95% CI, 9.8–10.1).

### 3.3. Anemia-Preventive Strategies and Burden of Anemia

The level of compliance reported by women is displayed in Table 2. In this study, only 12.1% of women met our criteria of being “fully compliant” with the anemia-preventive strategies, with the majority of our sample falling into the category of “somewhat compliant” (63.5%). The level of anemia at term ranged from 23.7% in the fully compliant group to 84.2% in the non-complaint group; more than 80% of women who were classified as being somewhat compliant or non-compliant displayed anemia at term (i.e., mean Hb < 10.0 g/dL). Analysis of variance (F = 72.2; *p*-value < 0.01) revealed significant differences in mean Hb levels. Post hoc comparisons revealed that mean Hb levels in the groups were significantly different among all compliance groups based on Bonferroni multiple comparison tests (*p* value < 0.001). 

The level of adherence to each of the four anemia-preventive strategies is summarized in Table 2. For prevention of malaria during pregnancy, ITN use in the past three months was commonly reported by participants (88.7%, 681/768). Less than one-quarter of women (21.5%, 165/768) reported taking antimalarial prophylaxis as per existing guidelines (at least two doses of SP) during pregnancy. Deworming with either mebendazole or albendazole was uncommon among women, with only 5.6% (43/768) reporting deworming at least twice over their pregnancy. Approximately, a quarter (26.9%) of women reported using iron and folate for at least 3 months or longer. For each anemia-preventive strategy, the mean Hb level of women who did not adhere to that strategy was less than 10.0 g/dL.

### 3.4. Factors Associated with Poor Compliance among Pregnant Women

The factors significantly associated with poor compliance in the bivariate and multivariate analyses are presented in Table 3. Maternal age, gravidity, and marital status were not statistically significant in the bivariate analysis. No formal or primary education level and delaying the first ANC visit to the second trimester were associated with poorer compliance to the anemia-preventive measures. Participants with no formal or primary level of education had two-times increased odds of not complying with anemia-preventive interventions compared to participants with secondary or tertiary education. Initiating ANC visits in the 2nd or 3rd trimester was also associated with increased odds of poorer compliance compared to women who initiated ANC visits in the 1st trimester.

### 3.5. Sub Analysis for Participants Whose Hb Levels Had Been Assessed at Initiation of ANC and at Term

In Table 4, a sub-analysis that included 564 participants who had Hb levels assessed at initiation of ANC and at term revealed that, on average, Hb levels remained relatively stable for the fully compliant and mainly compliant groups. However, in the somewhat compliant and noncompliant groups, there was a significant decrease in mean Hb levels from ANC initiation to term, which is likely explained by late initiation of first ANC visit either at 2nd or 3rd trimester as per Table 3. In Table 3, we observed that women who attended ANC services late (i.e., 2nd or 3rd trimester) were likely to fail to comply with anemia-preventive strategies.

### 3.6. Sensitivity Analysis

A sensitivity analysis was conducted to determine if changes to the classification score for compliance influenced mean Hb levels and anemia prevalence across levels of compliance. Compliance was re-classified into three groups. A score of 3–4 was classified as “fully/mainly compliant”, a score of 2 was classified as “somewhat compliant”, and scores of 0–1 were classified as “noncompliant”. Higher prevalence of anemia continued to be observed among women in the somewhat compliant and noncompliant groups, with mean Hb below 11 g/dL. See Table 5 below.

## 4. Discussion

Anemia is common among pregnant women in Tanzania and has been reported in more than half despite almost universal attendance to ANC clinics [6]. Although research has examined factors that may be associated with compliance of iron and folate supplements [25,27,29,30,31], this is a first study, to the best of our knowledge, that examines compliance across all four recommended anemia-preventive strategies using a composite outcome. These strategies include use of ITNs, antimalarial prophylaxis, medications for deworming, and iron and folate supplements. The study assessed the level of anemia at term, compliance with anemia-preventive strategies, and factors that were associated with compliance among women who delivered at health facilities in Mwanza city, Northwest Tanzania. In this study, we found that only 12.1% of pregnant women were reported to be “fully compliant” (i.e., reported to comply with all four anemia-preventive strategies). The prevalence of anemia among these women was 23.7%. In contrast, a much higher prevalence of anemia (84.2%) was observed among women who were classified as not complying with the preventive strategies. Initiating ANC in the 2nd or 3rd trimester and lower or no formal education were significantly associated with poor compliance. For women who were classified as fully compliant and mainly complaint, their Hb level from time of ANC initiation to delivery remained stable in contrast to women in the somewhat compliant and noncompliant groups, who displayed significantly lower Hb levels. Oh et al., in their review, reported anemia reduction by 47% for women using iron supplementation, which suggests that adherence to preventive strategies reduces the burden of anemia [20].

Anemia in pregnancy remains to be a public health concern. In this study, two-thirds of pregnant at-term women admitted for delivery were anemic despite the existence of preventive measures as stipulated in FANC model. This prevalence level was higher than the prevalence of anemia in pregnancy (57%) reported in Tanzania Demographic Health Survey (TDHS) report 2015/16 and by Liyew et al. in a secondary analysis of data from demographic health surveys from some of African countries [6,32]. The persisting burden of anemia is exaggerated by the physiological demands of pregnancy across trimesters [5]. The observed differences between the present study and previous research could be due to the fall in Hb concentrations that occurs across trimesters, as all of participants were in the present study were in their third trimester. Further, the observed higher prevalence of anemia at term aligns with the burden of anemia among women in the reproductive-age group [33].

Although many countries report universal ANC coverage, anemia persists and threatens maternal and child health, particularly in Africa [2,34,35]. In Tanzania, maternal deaths are estimated to be 595 per 100,000 live births, with primary hemorrhage ranking first as the cause of premature death among pregnant women [6,36,37]. The association between anemia and blood loss has been documented in studies, suggesting that anemic women are more likely to experience detrimental outcomes during delivery and post delivery [10,38,39]. The observed high prevalence of anemia in pregnant at-term women in Mwanza, Northwestern Tanzania, could attenuate efforts focused on reducing of maternal mortality.

It is standard clinical practice for all pregnant women to be prescribed and provided with iron supplements, malaria prophylaxis, and anthelminthic, as per the currently implemented FANC model in Tanzania [22]. During antenatal care visits, all pregnant women are provided with a free ITN and are encouraged to sleep under it. The integration of anemia-preventive strategies in ANC clinics is an attempt to reduce the prevalence of this condition in pregnant women [40]. Sleeping under an ITN can prevent malaria in pregnant women if it is appropriately used together with antimalarial prophylaxis [18]. We found that the use of ITNs among pregnant women was high at 88.7%, which could be attributed to the fact that ITNs are distributed free of charge to pregnant women during their first antenatal visit, and health providers encourage their use. However, only 21.5% of participants reported using an antimalarial at least twice during pregnancy, as recommended by WHO and the Tanzanian national guidelines [13,41]. Northwest Tanzania is a malaria-endemic area, and prevalence of malaria among pregnant women is estimated to be approximately 12.8% [42]. Furthermore, since the majority of participants attended an ANC clinic at least once, the provision of health education on the importance of using bed nets could partly account for the use of ITNs by a majority of pregnant women. However, the reported low use of antimalarials in our participants places them at risk from mosquito bites before retiring to bed, hence increasing their risk of contracting malaria. These findings differ from the TDHS report, which documented that ITNs were used by 54% of pregnant women, and at least two doses antimalarial were taken by 33% of pregnant women in the Lake zone, the region in which this study was conducted [6]. Differences in the findings could be due to sample selection and recall bias. In the present study, only women reported on their use of ITNs and antimalarial at term, whereas in the TDHS survey, participants were women who had had a live birth in the preceding two years. Therefore, recall biases in the latter study could have resulted in an underestimation of ITN use and an overestimation in the use of antimalarials due to time lag between the pregnancy and when the survey was undertaken.

Administration of deworming medications to pregnant women may also reduce anemia in worms-endemic areas [19], although the benefits have not been supported in randomized trials [43,44,45]. In this study, only a few women (5.6%) reported taking two doses of mebendazole/albendazole over the pregnancy period as recommended. This is a very low level of compliance compared to the national level of 63% reported in the TDHS 2015/16 and 33% in rural regions [6,35]. Again, it is of note that the TDHS included women who gave birth 2–5 years prior to the study, which may introduce recall bias, leading to overestimation of usage. Our study also found that only 18.9% of participants reported using iron and folate supplements for more than 3 months. This finding is consistent with those reported on the TDHS 2015/16 (i.e., 21.4%) [6].

We found that poor compliance to the anemia-preventive strategies (composite variable) was associated with late attendance to ANC and low education level. Early initiation of ANC remains sub-optimal in Tanzania, with only 24% of pregnant women attending ANC clinics before the 24th week of their pregnancy [6]. Initiation of ANC in the second or third trimester, however, could affect the health of the mother and the unborn baby, as prenatal conditions of concern may not be identified early (e.g., anemia, gestational diabetes, high blood pressure, chronic infections), and effective interventions may not be initiated in a timely fashion [46,47,48,49]. Furthermore, late initiation of ANC may result in women not being provided with the recommended ANC services, as health care workers may be unable to provide all of the interventions in remaining period of pregnancy.

To the best of our knowledge, no published study has investigated adherence to all four anemia-preventive strategies in LMICs. Notably, we used a composite variable that considered four anemia-preventive strategies, namely use of ITNs, anthelminthic, antimalarial prophylaxis, and iron and folate supplements. The information generated from this study highlights the lack of compliance to anemia-preventive strategies among pregnant women who attended health facilities at term in Mwanza, Northwest Tanzania. Poor compliance to preventive strategies could partially account for the persistence of maternal anemia and the high maternal mortality rate in Northwest Tanzania. However, the findings from this study should be interpreted with caution. Pill counts as a measure of adherence to iron and folate supplement and biomarkers for checking concentration in the blood were not carried out. As we used a self-report approach, which is compromised by social desirability bias, it is possible that women may have reported high levels of compliance. However, the fact that the mean Hb level and prevalence of anemia varied significantly across the levels of compliance gives us greater confidence in the maternal self-reports. Another additional limitation is that the study was based on a selected sample of women who delivered at term and who chose to deliver at a formal health facility. For those who delivered their newborn at home, we may possibly uncovered different findings. Lastly, we did not account for the iron-rich food a woman consumed, which may reduce the likelihood of anemia, which in turn may influence the likelihood of the woman complying with the ANC strategies.

## 5. Conclusions

Anemia remains a public health problem in pregnancy in LMIC, with observed poor compliance with the preventive strategies recommended by the WHO. To obtain the full benefits of ANC and to reduce maternal anemia, women should be encouraged to initiate ANC during first trimester of pregnancy. Further research is recommended to explore possible reasons for poor compliance in the use of antimalarial prophylaxis, deworming medication, and iron and folate supplements. Future studies can further distinguish pregnant women who were non-complaint compared to the women who complied with one of the anemia-preventive strategies, namely either anti-helminthic therapy or antimalarial prophylaxis.

## Figures and Tables

**Table 1 nutrients-14-03850-t001:** Socio-demographic and obstetric characteristics of pregnant women at term from Mwanza city, Tanzania.

Patient Characteristic	Number of Women (*N* = 768)	Percent (%)
Residence		
Nyamagana	339	44.1
Ilemela	354	46.1
Other	75	9.8
Marital status		
Currently single	119	15.6
Currently married	649	84.4
Level of Education		
None	51	6.6
Primary	412	53.7
Secondary	241	31.4
College and above	64	8.3
Economic activity		
Subsistence farmer	111	14.5
Housewife	313	40.8
Small scale trader	257	33.5
Gainfully employed	74	9.5
Other	13	1.7
Religion		
Christian	622	81.0
Muslim	139	18.1
Other	7	0.9
Gravidity		
1	285	37.1
2	175	22.8
≥3	308	40.1
Gestation trimester at initial ANC visit		
First trimester	170	22.1
Second trimester	526	68.5
Third trimester	72	9.4
Number of ANC visits		
None	2	0.2
1–3	334	43.5
≥4	432	56.3
Health facility attended		
Dispensary	480	62.5
Health center	177	23.1
Hospital	111	14.4

**Table 2 nutrients-14-03850-t002:** Anemia preventive strategies among 768 pregnant women in Mwanza, Tanzania.

Specific Preventive Strategy	*N* (%)	Anemia Prevalence % or (95% CI)	Mean Hb (SD) or (95% CI)
^a^ ITN use for past 3 months			
No	87 (11.3)	75.9	9.6 (2.3)
Yes	681 (88.7)	67.8	10.0 (1.9)
^b^ Malaria prophylaxis			
Reported < 2 doses	603 (78.5)	75.9	9.7 (1.9)
Completed ≥ 2 doses	165 (21.5)	42.4	11.1 (1.9)
^c^ Deworming medications			
Reported < 2 doses	725 (94.4)	70.2	9.9 (2.0)
Completed ≥ 2 doses	43 (5.6)	44.2	10.9 (1.9)
^d^ Iron and folate supplements			
Reported for < 3 months	623 (81.1)	78.5	9.6 (1.9)
Reported for ≥ 3 months	145 (18.9)	26.9	11.6 (1.6)
Compliance level			
Fully compliant	93 (12.1)	23.7 (16.1,33.4)	11.7 (11.4,12.1)
Mainly compliant	130 (16.9)	50.0 (41.4,58.6)	10.8 (10.5,11.1)
Somewhat compliant	488 (63.5)	80.5 (76.8,83.8)	9.5 (9.4,9.7)
Noncompliant	57 (7.4)	84.2 (72.2,91.6)	9.1 (8.5,9.7)

^a^ ITN, insecticide-treated net. ^b^ At least two doses of antimalaria were considered as complete doses as per ANC guidelines. ^c^ At least two doses of deworming medications were considered as complete doses as per ANC guidelines. ^d^ Taking iron and folate supplements for more than 3 months or more was considered an anemia-preventative strategy.

**Table 3 nutrients-14-03850-t003:** Factors associated with poorer compliance to anemia-preventive strategies among study participants.

Characteristic	Bivariate Analysis		Multivariate Analysis	
	OR	95% Confidence Interval	OR	95% Confidence Interval
Maternal age	1.01	0.98–1.03		
Currently married				
No	1			
Yes	1.08	0.73, 1.61		
Education level				
Secondary or Tertiary	1		1	
Primary and None	2.03	1.51, 2.73	2.04	1.52, 2.74
Trimester at 1st ANC visit				
1st trimester	1		1	
2nd trimester	1.77	1.27, 2.47	1.78	1.27, 2.50
Use of Herbal Medicines				
No	1			
Yes	1.17	0.81, 1.69		

**Table 4 nutrients-14-03850-t004:** Change in mean Hb for each level of compliance among 564 participants.

Compliance Level	Number of Participants	Mean Hb at ANC Initiation	Mean Hb at Term	Mean Diff	95% CI for Diff	*p*-Value
Fully compliant	75	11.76	12.04	−0.28	−0.65, 0.09	0.14
Mainly compliant	99	11.09	10.80	0.29	−0.10, 0.68	0.14
Somewhat compliant	351	10.13	9.58	0.55	0.36, 0.73	0.00
Not compliant	39	10.14	9.37	0.77	0.16, 1.38	0.01

**Table 5 nutrients-14-03850-t005:** Prevalence of anemia and mean Hb levels in sensitivity analysis.

Compliance Level	Number of Participants *n* (%)	Prevalence % (95% CI)	Mean Hb (g/dL) (95% CI)
Noncompliant	545 (70.96)	80.92 (77.39, 84.00)	9.45 (9.34, 9.66)
Somewhat compliant	130 (16.93)	50.00 (41.44, 58.56)	10.78 (10.46, 11.09)
Fully compliant	93 (12.11)	23.66 (16.07, 33.40)	11.74 (11.41, 12.07)

## Data Availability

Data is available through a request to corresponding author.

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
