# Peer review of "High Prevalence of Anemia and Poor Compliance with Preventive Strategies among Pregnant Women in Mwanza City, Northwest Tanzania: A Hospital-Based Cross-Sectional Study"

_nutrients, 2022, doi:10.3390/nu14183850_

Round 1
Reviewer 1 Report
the manuscript is clear and generally well written. To my opinion, it can be accepted in the present form.
Author Response
Thank you so much for taking time to review our work
Reviewer 2 Report
This is an interesting and carefully performed study. It demonstrates a clear association between compliance with a four component maternal anemia prevention program and anemia at term. They demonstrate that complete compliance with all four components only occurs in 12% of patients, but that complete compliance with any three of the four elements produces a satisfactory outcome. They also demonstrate maternal characteristics, mostly related to education, that contribute to failure to comply and that would provide a reasonable target for focused intervention in the maternal population.
The authors make clear that they consider compliance with three elements to be full compliance. However, in the discussion, they need to re-emphasize that for the purposes of their analysis, complete compliance with three of the four elements was considered full compliance.
For a future study, the authors may wish to look at a subdivision of their patients who score “0” on one of the components and see whether patients who, for example, took part of the anti-helminthic therapy are distinguishable from the patients who took none of it.
Author Response
Thank you so much for your inputs.
We have emphasized on the composite outcome (compliance) in discussion as you suggested.
We also considered your suggestion of writing another manuscript that will explore the difference between those who complied vs non complaints. Thank you so much.